# Transformers Can Learn Connectivity in Some Graphs but Not Others

## Abstract

Highly competent reasoning capability is essential to ensure the factual correctness of the responses of transformer-based Large Language Models (LLMs), and robust reasoning about transitive relations is instrumental in many settings, such as causal inference. Therefore, it is essential to investigate the capability of transformers in the task of inferring transitive relations (e.g., knowing A causes B and B causes C, we can infer that A causes C). The task of inferring transitive relations is *equivalent* to the task of connectivity in directed graphs (e.g., knowing there is a path from A to B, and there is a path from B to C, we can infer that there is a path from A to C). Past research focused on whether transformers can learn to infer transitivity from in-context examples provided in the input prompt. However, transformers' capability to infer transitive relations from training examples and how scaling affects this ability is unexplored. In this study, we endeavor to answer this question by generating directed graphs to train transformer models of varying sizes and evaluate their ability to infer transitive relations for various graph sizes. Our findings suggest that transformers are capable of learning connectivity on "grid-like" directed graphs where each node can be embedded in a low-dimensional subspace, and connectivity is easily inferable from the embeddings of the nodes. We find that the dimensionality of the underlying grid graph is a strong predictor of transformers' ability to learn the connectivity task, where higher-dimensional grid graphs pose a greater challenge than low-dimensional grid graphs. In addition, we observe that increasing the model scale leads to increasingly better generalization to infer connectivity over grid graphs. However, if the graph is not a grid graph and contains many disconnected components, transformers struggle to learn the connectivity task, especially when the number of components is large. We also find that transformers benefit more from increasing the graph size than increasing the model size. The code of our experiments is publicly available at github.com/anonymoususer437/transformers_graph_connectivity.

## 1 Introduction

Transformer-based Large Language Models (LLMs), with their impressive generative capabilities, are widely adopted in various generative AI applications. Improving the logical reasoning capabilities of LLMs is essential for competent performance on complex reasoning problems. Transitivity is a fundamental property of many real-world relations (e.g., knowing A causes B and B causes C, we can infer that A causes C). Reasoning over transitive relations is equivalent to many other reasoning tasks such as graph connectivity (Fatemi et al., 2024), question answering (Murphy et al., 2025), deductive reasoning (Saparov et al., 2025; Hoppe et al., 2025), mathematical reasoning (Trinh & Luong, 2024), logical reasoning (Sullivan & Elsayed, 2024; Gaur & Saunshi, 2024), temporal reasoning about multiple events (Fatemi et al., 2025), program analysis (Ceka et al., 2025) as well as causal reasoning (Vashishtha et al., 2025; Jin et al., 2023; 2024). Existing studies primarily focus on augmenting the ability of LLMs to reason about transitive relations using in-context examples (i.e., about information given in the prompt; Vashishtha et al., 2025). However, LLMs learn vast quantities of information from the pre-training corpus, and whether transformers (Vaswani et al., 2017; Joshi, 2025) are able to reason about transitive relations from examples given during pretraining is relatively unexplored.

 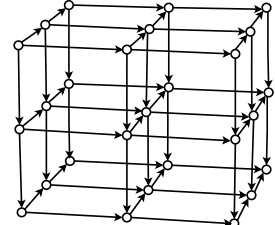

Figure 1: Examples of 2 and 3-dimensional grid graphs with 8 and 27 nodes, respectively.

To assess the capability of transformers in learning logical reasoning from training examples generally, it is essential to investigate their ability to infer transitive relations. This is equivalent to learning connectivity or path-finding in directed graphs. The problem of connectivity between graph nodes has been studied in recent work (Vashishtha et al., 2025; Sanford et al., 2024). Reasoning over transitive relations is an instrumental capability in many reasoning tasks, such as inferring causation in causal reasoning. In causal discovery, the edges of a directed graph denote causal relationships, in which case, finding a path in this graph is equivalent to inferring a causal relationship between two events (Sheth et al., 2025; Joshi et al., 2024). In Sheth et al. (2025), the edges of the directed graph are provided as in-context examples in the input prompt of LLMs with a path-finding query (i.e., "*A causes B, B causes C. Does A cause C?*"). However, it is crucial to investigate the ability of transformers to infer transitive relations from training, as the LLM heavily relies on knowledge acquired during pretraining for downstream applications. Some recent studies (Zečević et al., 2023; Zhang et al., 2024) claim that transformers learn to memorize patterns without understanding the logical connections between entities (e.g., if "lightning" appears often in the context of "thunder" in the training corpus, transformers will likely learn that there is a causal relation between the two entities, even if no such relation exists). Keeping this pattern-learning phenomenon of transformers in mind, it is essential to investigate the extent to which transformers can generalize when reasoning about transitivity.

In this work, we aim to understand whether transformers can learn to infer transitivity, and equivalently, whether they can learn connectivity over directed graphs when connectivity information between node pairs is provided as individual training examples during pre-training. We hypothesize that transformers are more likely to learn to infer connectivity over graphs whose nodes are embeddable into a low-dimensional subspace such that connectivity can be easily inferred from the embedding of each node. As such, we define the notion of a *grid graph*. A one-dimensional grid graph with $n$ nodes can be defined as a chain with $n$ nodes, where the nodes are defined as points in a line with all edges directed in a single direction. A grid graph with dimensionality $k$ can be viewed as a $k$-dimensional grid where nodes are defined as grid points and the edges exist between adjacent grid points in a single direction for each dimension. In Figure 1, we show examples of two-dimensional and three-dimensional grid graphs with 8 and 27 nodes, respectively. Disconnected graphs, on the other hand, are not as easily embeddable into such a low-dimensional subspace where connectivity can be easily inferred. Therefore, we will explore whether transformers can learn connectivity in grid graphs more easily as compared to disconnected chain graphs, which contain multiple components where each component is a connected chain.

Our experimental analysis indicates that transformers are more successful in learning to infer connectivity on grid graphs and not always successful on disconnected chain graphs. The existence/nonexistence of a path from the low-dimensional vector representation of nodes helps transformers to infer the connectivity over grid graphs. To understand how well transformers perform reasoning over transitive relations at scale, we study how scaling the model size and the graph size (i.e., number of nodes) impacts the ability of transformers to infer transitive relations for grid graphs and disconnected chain graphs. Our experimental results suggest that scaling the model size facilitates learning to infer transitive relations over grid graphs, while comparable improvement is not observed for disconnected chain graphs. When we scale the graph size, performance remains consistent for both small and large grid graphs. Although transformers struggle to learn connectivity for smaller disconnected chain graphs, consistent improvement is observed for larger disconnected chain graphs with a fixed number of components (i.e., chains). In addition, transformers show better performance for grid graphs with lower dimensionality as compared to higher dimensionality, and increasing the number of nodes in grid graphs helps transformers to learn connectivity on

higher-dimensional grid graphs. Similarly, for disconnected chain graphs, transformers show better performance for a small number of chains, for a fixed number of nodes, and transformers benefit more from scaling the graph size than model size in learning connectivity for disconnected chain graphs with a large number of chains.

**Research Questions:** In this work, we aim to answer the following research questions:

- **RQ1:** How well do transformers learn the connectivity task from training examples for grid graphs and disconnected chain graphs?
- **RQ2:** How does scaling the model size and graph size impact the performance of transformers to learn connectivity for grid graphs and disconnected chain graphs?
- **RQ3:** How does the grid dimension and the number of chains impact the performance of transformers to learn connectivity for grid graphs and chain graphs, respectively?

**Contributions:** The key contributions/findings of this work are as follows:

- We investigate the ability of transformers to *learn to* infer transitive relations, by evaluating their ability to infer transitivity, and equivalently, to infer connectivity on directed graphs, where a pair of nodes and the connectivity information are provided as individual training examples, instead of in-context examples in the prompt.
- Transformers excel at learning connectivity in low-dimensional grid graphs, as it's ability deteriorates with increasing dimensionality of the underlying grid graph, while scaling graph size helps more for higher-dimensional grid graphs as compared to scaling model size.
- Scaling the model size makes it easier for transformers to learn connectivity over grid graphs as compared to disconnected chain graphs, while scaling the graph size, transformers' performance stays consistent on grid graphs and improves for disconnected chain graphs with a large number of nodes.
- Transformers struggle to learn connectivity for higher-dimensional grid graphs and disconnected chain graphs with a large number of chains, and transformers benefit more from scaling the graph size than scaling the model size in both settings.

## 2 RELATED WORK

**Language Models for Transitivity/Graph Connectivity**: To determine the reasoning abilities of the transformer architecture, theoretical studies have been conducted, which suggest that a logarithmic number of layers is necessary and sufficient to learn the connectivity problem for graphs (Sanford et al., 2024). To feed the graph into the transformer architecture, several studies (Fatemi et al., 2024; Vashishtha et al., 2025) provide the edges of the graph in the prompt and ask questions about the connectivity of the graph. Providing graph edges in the prompt assumes that transformers themselves have the capability to reason over the graph structure. However, some recent work shows that transformers perform pattern memorization in data rather than learning transitive relationships (Joshi et al., 2024; Zečević et al., 2023). Recently, Saparov et al. (2025) mentioned that transformers struggle to perform depth-first search in graphs. For larger graphs, the context length of LLMs places an upper bound on the size of in-context graphs. Therefore, it is important to study the problem of learning connectivity or finding paths in directed graphs by training transformers to infer transitivity/connectivity between entities, providing the connectivity information as individual training examples.

**Scaling Laws for Language Models:** Increasing the size of the model or data and observing the impact on model performance is essential to understand how scaling impacts the performance of a model on a particular task (Kaplan et al., 2022; Finzi et al., 2025; Qin et al., 2025; Wu et al., 2025). Kaplan et al. (2022) suggest that scaling the model size, data size or amount of compute used for training has a greater effect versus other variations in the architecture such as number of layers. Therefore, we study how scaling the model size, data size, and amount of training compute change the performance of transformers on the connectivity task for directed graphs. To identify the optimal size of the model, scaling experiments have been conducted with language models to predict part of the knowledge graph triples (Wang et al., 2025). Hence, it is necessary to investigate how scaling impacts the capabilities of transformers on fundamental tasks such as graph connectivity. The dependence of scaling curves on data complexity has also been covered in recent research (Pandey, 2024; Yin et al., 2024). Furthermore, Roberts et al. (2025) corroborate the observation that model scaling has a greater impact on knowledge-intensive downstream tasks, while data scaling has a greater im-

pact on reasoning-intensive tasks. This motivates us to explore how model and data scaling impact the performance on the connectivity task for different topologies of directed graphs.

# 3 GRAPH CONNECTIVITY AND REASONING

In this section, we describe the task that we consider in our study.

**Definition 1** (*Connectivity in Directed Graphs*). *Given a directed graph $G = (V, E)$, where $V$ and $E$ denote the set of nodes and directed edges, respectively, for any two distinct nodes $v_{start}, v_{goal} \in V$ where $v_{start} \neq v_{goal}$, we define the **connectivity function** $\mathcal{T}(v_{start}, v_{goal})$ as the indicator function that denotes whether there is a path from $v_{start}$ to $v_{goal}$. That is, $\mathcal{T}(v_{start}, v_{goal}) = 1$ if and only if there exists a simple path from $v_{start}$ to $v_{goal}$ using the directed edges in $E$. Otherwise, $\mathcal{T}(v_{start}, v_{goal}) = 0$. For brevity, we will refer to the task of predicting the connectivity function as the **connectivity task**.*

The connectivity task is equivalent to the task of inferring transitive relations. More precisely, consider a relation $r$ (e.g., causes) and a set of facts of the form $r(x, y)$ (e.g., causes(sunlight,photosynthesis), causes(photosynthesis, oxygen)). We say the relation $r$ is *transitive* if for any $x, y, z$, if $r(x, y)$ and $r(y, z)$, then $r(x, z)$. Given a starting concept $x_{start}$ (e.g., sunlight) and a goal concept $x_{goal}$ (e.g., oxygen), the task of inferring transitive relations is to determine whether $r(x_{start}, x_{goal})$ (e.g., causes(sunlight,oxygen)) is provable from the given set of facts. It is straightforward to draw a one-to-one correspondence between examples of the connectivity task and examples of the task of inferring transitive relations: First define a map $\mathcal{M}$ between concepts and nodes, then for any fact $r(x, y)$, we create a directed edge from $\mathcal{M}(x)$ to $\mathcal{M}(y)$. The task of finding whether $r(x_{start}, x_{goal})$ is true is equivalent to finding whether there is a path from $\mathcal{M}(x_{start})$ to $\mathcal{M}(x_{goal})$.

The connectivity task is also equivalent to deductive reasoning in a simplified logic called *implicational logic*. In this logic, all logical forms have the form $A \rightarrow B$ (i.e., "if $A$, then $B$"). Thus, given a set of facts of the form $A \rightarrow B$, a premise $X_{start}$ and a goal $X_{goal}$, the deductive reasoning task is to determine whether $X_{goal}$ is provable. For example, given the facts $\forall x(\text{panda}(x) \rightarrow \text{bear}(x))$ (i.e., "all pandas are bears"), $\forall x(\text{bear}(x) \rightarrow \text{furry}(x))$ ("all bears are furry"), and panda(po) (i.e., "Po is a panda"), we can deduce furry(po) ("Po is furry"). It is similarly straightforward to construct a one-to-one equivalence between reasoning problems in implicational logic and examples of the connectivity task.

## 3.1 LEARNING CONNECTIVITY AT TRAIN-TIME

To train a transformer to perform the connectivity task on a directed graph $G = (V, E)$, we first define the vocabulary as $\mathcal{V} = V \cup \{\text{"Y"},\text{"N"}\}$ (we assume a simple tokenization scheme where each node is mapped to a single unique token). The training data consists of a large set of examples, where each example consists of a pair of nodes $v_{start}$ and $v_{goal}$ with a corresponding label, "Y" or "N", indicating whether there exists a path from $v_{start}$ to $v_{goal}$. The nodes $v_{start}$ and $v_{goal}$ form the input tokens to the transformer architecture, while the "Y" or "N" label is the ground-truth prediction. We train the transformer architecture using the cross-entropy loss on the output token.

## 3.2 CONNECTIVITY OVER GRID GRAPHS

We aim to describe a class of graphs for which it is easy for transformers to learn connectivity.

**Definition 2** (*Grid graph*). *Given a graph $G = (V, E)$, we say that $G$ is a $k$-**dimensional grid graph** if there exists an embedding $\Psi : V \mapsto \mathbb{R}^k$ such that: For any two nodes $u, v \in V$, there exists a path from $u$ to $v$ if and only if the difference between the embedding of $v$ and that of $u$ contains no negative elements (i.e., $[\Psi(v) - \Psi(u)]_i \geq 0$ for all elements $i \in \{1, \ldots, k\}$).*

Without loss of generality, we may assume that the nodes of a $k$-dimensional grid graph can be represented as a $k$-dimensional vector of positive integers. If we carefully observe the nodes of $k$-dimensional grid graph, we can observe that if there is a path from $v_{start}$ to $v_{end}$ then $\Psi(v_{end}) - \Psi(v_{start})$ will have non-negative entries in all $k$ dimensions, If there is no path from $v_{start}$ to $v_{end}$

then there will be at least one negative entry in $\Psi(v_{end}) - \Psi(v_{start})$. We present an example of a grid graph with dimensionality $k = 2$ and 4 nodes, and an example of an embedding $\Psi$ in Figure 12 and Table 13 in the Appendix.

Because of this pattern, for a $k$-dimensional grid graph, the existence/nonexistence of a path can be determined by the difference of the $k$-dimensional embeddings of the endpoints, and therefore, if a model is able to learn the embedding $\Psi$ on a graph $G$, it would effectively learn to perfectly solve the connectivity task on $G$. Thus, we hypothesize that for small values of $k$, transformers more easily learn the connectivity task on $k$-dimensional grid graphs.

We contrast the notion of a grid graph with that of disconnected chain graphs, which are not as easily embeddable in low-dimensional subspaces. For example, consider the graph containing the two components $A \rightarrow B \rightarrow C$ and $D \rightarrow E \rightarrow F$. Each component is a 1-dimensional grid graph, but the nodes are not embeddable in one dimension such that pairwise connectivity is easily inferable from comparisons of the embeddings (e.g., there is no function $\Psi$ such that $\Psi(v) \geq \Psi(u)$ if and only if $v$ is reachable from $u$, for any two nodes $u, v$). We experiment training transformers on grid graphs and disconnected chain graphs and empirically measure how easily the model learns the connectivity task in each case.

## 4 EXPERIMENTS

In this section, we perform experimental analysis to understand how transformers can learn the connectivity task over grid graphs and disconnected chain graphs when we scale the model size and the graph size. We additionally aim to explore how the grid dimension and the number of chains affects learning dynamics and scaling behavior.

### 4.1 EXPERIMENTAL SETUP

**Experimental Settings:** For our experiments, we first generate a grid graph with number of nodes $n$ and grid dimensionality $k$. In addition, we generate a disconnected chain graph with parameters: total nodes $n$, number of chains $C$, with chain length $L = \lfloor \frac{n}{C} \rfloor$[1] . For each generated graph, we produce connectivity examples: For each example, we sample a pair of nodes $(v_{start}, v_{end})$, which forms the two-token input to the transformer, and we utilize their connectivity as the ground truth label denoted as "Y" or "N" (indicating the presence or absence of a path from $v_{start}$ to $v_{end}$). We use learned token embeddings with dimension $d_{emb}$, initialized randomly. Also, we concatenate absolute positional encoding of length $d_{pos} = 2$ for each token with the token embedding and an additional hidden dimension of $d_{hid} = 32$ in our experiments. By concatenating the token embedding, positional encoding, and hidden dimensions, we obtain the input embedding for the transformer model which has dimension $d_{model} = d_{emb} + d_{pos} + d_{hid}$. We experiment with varying $d_{emb}$ and observe the effect on transformers learning the connectivity task. We set the feed-forward dimension $d_{ff} = d_{model}$. In addition, we use pre-layer normalization as in Xiong et al. (2020), set dropout to 0, and the number of layers $l = 4$. During the experiments, we record the accuracy and loss of the model as a function of training compute, in terms of floating point operations (FLOPs). We report the loss and accuracy for both training and test sets on the $y$-axis and FLOPs on the $x$-axis. We choose to measure model performance vs FLOPs as opposed to epochs, since for different model/graph sizes, the number of FLOPs for each iteration is not constant. We use a logarithmic scale with base 10 in the plots for loss and FLOPs for better visualization. For grid graphs, we experiment with $k = 2$-dimensional grid graphs with number of nodes $n = 50, 100, 200, 400, 800$. For the disconnected chain graphs, we set the number of chains $C = 10$ and number of nodes $n = 50, 100, 200, 400, 800$. To determine the impact of grid dimension and the number of chains, we experiment with grid dimension $k = 1, 2, 3, 4$ and $5$ and number of chains $C = 1, 5, 10, 15$ and $20$. We chose these parameters to make the number of nodes maximally comparable between grid graphs and disconnected chain graphs. For all experiments, we run with ten different random seeds, and plot the mean and standard deviation in all figures.

**Grid Graph Generation:** To generate a $k$-dimensional grid graph with $n$ nodes with node IDs $1, \ldots, n$, we first compute the grid width $b = ceil(n^{\frac{1}{k}})$. We convert the ID of each node into a

---

[1]If $n$ is not divisible by $C$, the remainder nodes are formed into an additional chain.

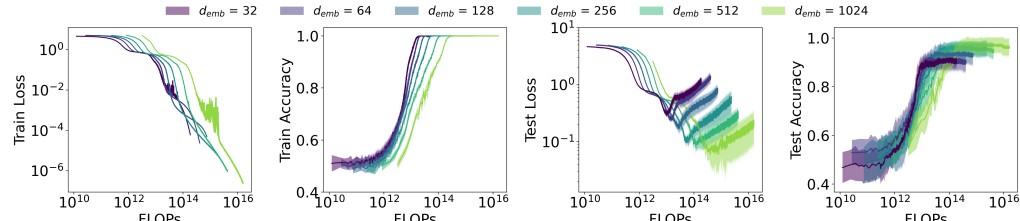

Figure 2: Accuracy and loss vs. training compute on the connectivity task for a grid graph with number of nodes $n = 100$ and grid dimension $k = 2$, for transformers of various sizes/model dimensions with $d_{emb} = 32, 64, 128, 256, 512$ and $1024$.

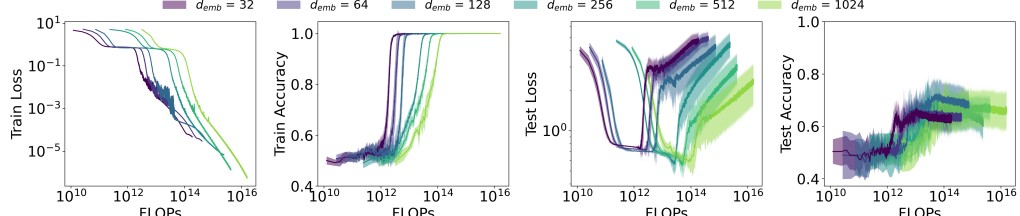

Figure 3: Accuracy and loss vs. training compute on the connectivity task for a disconnected chain graph with number of nodes $n = 100$ and number of chains $C = 10$, with $d_{emb} = 32, 64, 128, 256, 512$ and $1024$.

number in base-$b$ and take the digits to form a $k$-dimensional vector (i.e., in $\mathbb{Z}^k$). For any two nodes $u, v$, we add an edge $u \rightarrow v$ if the difference between the vectors, $\Psi(v) - \Psi(u)$ is a one-hot vector.

**Chain Graph Generation:** To generate each disconnected chain graph, we consider two parameters: The number of nodes $n$ and number of chains $C \ll n$. We compute chain length $L = \lfloor \frac{n}{C} \rfloor$ and generate $C$ chains of length $L$. In the Appendix, we present pseudocode to generate grid graphs and disconnected chain graphs in Algorithm 1 and Algorithm 2, respectively.

**Training and Test Set Generation:** We define $(v, u)$ as a *reverse negative node pair* if there is a path from $u$ to $v$. If there is no path from $u$ to $v$ or from $v$ to $u$, we define $(v, u)$ as a *disconnected negative node pair*. To generate train and test sets of node pairs, we first add all pairs of vertices $(u, v)$ to the train set. To produce a test set, we randomly sample 40 positive pairs $(u, v)$ from the train set (i.e., $v$ is reachable from $u$) and add their reverse negative pairs $(v, u)$ without replacement to the test set. We repeat this for 40 disconnected negative node pairs. We add details of generating train and test sets in Algorithm 3 in the Appendix.

### 4.2 LEARNING CONNECTIVITY ON GRID GRAPHS VS. DISCONNECTED CHAIN GRAPHS

To determine how well transformers learn the connectivity task for grid graphs and disconnected chain graphs, we train the model for both types of graphs and record the training loss, training accuracy, test loss and test accuracy on the y-axis over the number of training FLOPs on the x-axis in Figures 2 and 3 for grid graphs and disconnected chain graphs, respectively. While for the grid graphs in Figure 2, we achieve low test loss, the test loss continues to diverge for the disconnected chain graphs in Figure 3. Consequently, transformers achieve near-perfect accuracy for grid graphs and struggle to generalize for disconnected chain graphs. Intuitively, for disconnected chain graphs transformers likely struggle as there exists no simple embedding of the graph into a low-dimensional subspace where connectivity is more easily inferable. However, the results support the hypothesis that transformers can more easily learn a low-dimensional embedding of a grid graph to predict the existence/nonexistence of paths between its nodes.

### 4.3 THE EFFECT OF MODEL SCALE

In Figures 2 and 3, we present results on the connectivity task with $n = 100$ nodes for grid graphs and disconnected chain graphs, respectively. In this experiment, we scale the model size by increasing $d_{emb} = 32, 64, 128, 256, 512, 1024$. Since smaller models use less compute for each training iteration, we run the models for 15000 epochs for $d_{emb} \in \{32, 64\}$, 10000 epochs for $d_{emb} \in \{128, 256\}$, and 5000 epochs for $d_{emb} \in \{512, 1024\}$. For both grid graphs and disconnected chain graphs, as the training compute increases, the training loss drops, approaching near

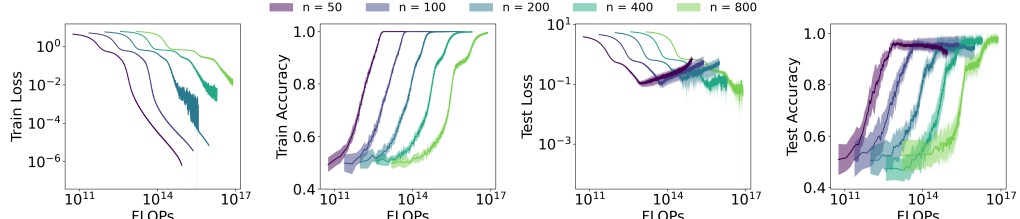

Figure 4: Accuracy and loss vs. training compute on the connectivity task for a grid graph with various graph sizes containing number of nodes $n = 50, 100, 200, 400, 800$ and grid dimension $k = 2$ with $d_{emb} = 256$.

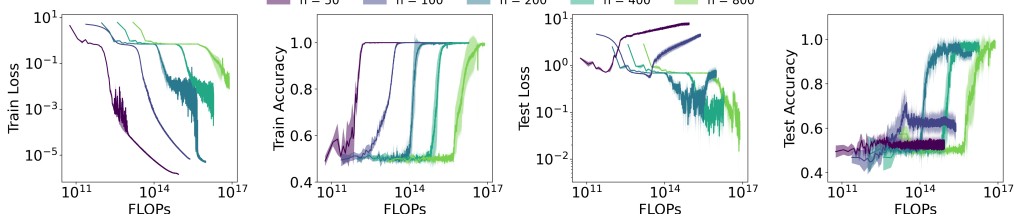

Figure 5: Accuracy and loss vs. training compute on the connectivity task for a disconnected chain graph with various graph sizes containing number of nodes $n = 50, 100, 200, 400, 800$ and number of chains $C = 10$ with $d_{emb} = 256$.

100% training accuracy. For the test loss curve, we can observe that increasing the model size leads to a lower test loss for grid graphs, and consequently, the test accuracy approaches 100%. However, for the case of disconnected chain graphs, increasing $d_{emb}$ instead increases the test loss. As a result, the test accuracy does not improve as we scale the model size for disconnected chain graphs. Therefore, increasing the model size gradually helps transformers to better generalize on the connectivity task over grid graphs; the same behavior is not observed for disconnected chain graphs. This suggests that, as the model size increases, transformers can better learn connectivity on grid graphs by utilizing a low-dimensional vertex-embedding heuristic, resulting in better generalization. But this phenomenon is not observed for the case of disconnected chain graphs.

### 4.4 THE EFFECT OF GRAPH SIZE

In Figures 4 and 5, we present results of training on the connectivity task for $k = 2$-dimensional grid graphs and chain graphs with number of chains $C = 10$ and number of nodes $n = 50, 100, 200, 400, 800$ with a fixed token embedding dimension $d_{emb} = 256$ for $15000, 10000, 10000, 5000, 5000$ epochs, respectively, to to enable fair comparison. In this experiment, we endeavor to measure how well transformers can learn the connectivity task when we increase the graph size by increasing the number of nodes in a graph. From the test loss and test accuracy curve in Figure 4, we observe that as we scale the graph size, the test loss slightly decreases and the test accuracy approaches 100%. Transformers' performance remains consistent as we increase the total number of nodes in grid graphs. As the number of nodes in grid graphs increases, the number of training pairs also increases, which provides more training data for transformers to learn the connectivity task by learning the difference between the vector representations of node pairs on grid graphs. On the other hand, for the case of the disconnected chain graphs, we can observe from Figure 5 that for graphs with a small number of nodes ($n = 50$ or $100$), test accuracy stays close to 60% and 80%, respectively. However, when we increase the number of nodes to $n = 200, 400, 800$, the test accuracy reaches close to 100%. For smaller graphs (e.g. 50 or 100 nodes) with a fixed number of chains $C = 10$, there are fewer nodes in each chain. This doesn't provide enough training data for transformers to learn graph connectivity for disconnected chain graphs. However, when we increase the number of nodes for large disconnected chain graphs by fixing the number of chains, it increases the number of nodes in each chain. As a result, transformers have more examples of connected node pairs, which helps to generalize better for the graph connectivity task.

### 4.5 THE EFFECT OF GRID DIMENSIONALITY

To evaluate how transformers learn connectivity for higher-dimensional grid graphs, we present the training dynamics for $k = 1, 2, 3, 4$ and 5-dimensional grid graphs in Figures 6, 7 and 8. First, we show results with number of nodes $n = 100$ and $d_{emb} = 256$ in Figure 6. Next, we increase the

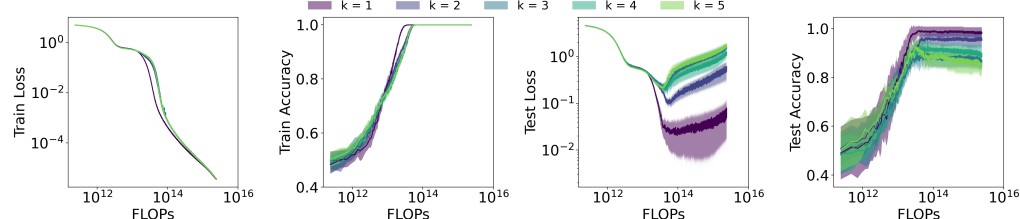

Figure 6: Accuracy and loss vs. training compute on the connectivity task for a grid graph with grid dimension, $k = 1, 2, 3, 4$, and $5$ and number of nodes $n = 100$ with $d_{emb} = 256$.

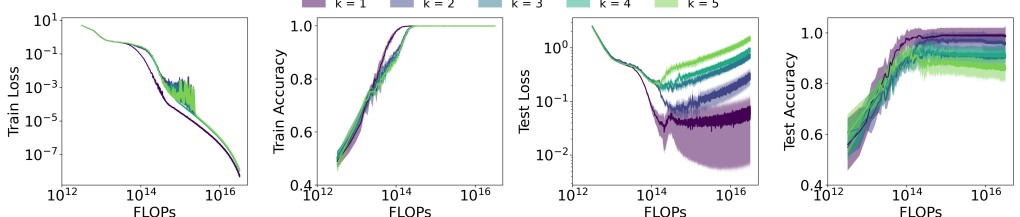

Figure 7: Accuracy and loss vs. training compute on the connectivity task for a grid graph with grid dimension, $k = 1, 2, 3, 4$, and $5$ and number of nodes $n = 100$ with $d_{emb} = 1024$.

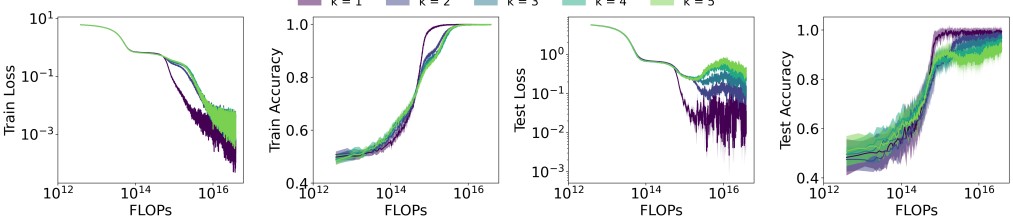

Figure 8: Accuracy and loss vs. training compute on the connectivity task for a grid graph with grid dimension, $k = 1, 2, 3, 4$, and $5$ and number of nodes $n = 400$ with $d_{emb} = 256$.

model size to $d_{emb} = 1024$, keeping the graph size unchanged, and show the result in Figure 7. Afterwards, in Figure 8 we increase $n$ to $400$, keeping the model size unchanged from Figure 6. From Figure 6, we observe that the test loss is lower for $1$-dimensional grid graphs as compared to $2, 3, 4$ and $5$-dimensional grid graphs. As a result, the test accuracy for $1$-dimensional grid graphs in Figure 6 is slightly higher as compared to higher-dimensional grid graphs, and much higher than $5$-dimensional grids. Next, when we scale the model size by increasing $d_{emb} = 1024$ and with the number of nodes constant at $n = 100$ in Figure 7, we can observe that transformers' performance for higher grid dimensionality in terms of test accuracy doesn't improve much. However, when we increase the number of nodes in the grid graph to $n = 400$ keeping the model size at $d_{emb} = 256$ in Figure 8, we observe that the test loss seems to decrease and test accuracy approaches closer to $100\%$ accuracy for higher grid dimensionalities, as compared to the case where $n = 100$ in Figures 6 and 7. This suggests that increasing the number of nodes in grid graphs is more effective than increasing the model size to better learn connectivity for higher-dimensional grids. When we increase the number of nodes in a grid graph, it provides more training data for transformers to better learn the low-dimensional embedding heuristic, whereas increasing the model scale does not provide as much improvement.

### 4.6 THE EFFECT OF THE NUMBER OF DISCONNECTED COMPONENTS

In Figure 9, we present the training dynamics for chain graphs with the number of nodes $n = 100$ for the number of chains $C = 1, 5, 10, 15, 20$ with $d_{emb} = 256$. We can observe that the test loss is comparatively lower when the number of chains is small (e.g., $C = 1$ or $5$) and although the test accuracy stays close to $100\%$ when the number of chains is small (e.g., $C = 1$ or $5$), it drops gradually as the number of chains increases (e.g., $C = 10, 15, 20$), akin to phase transition. For a fixed-size graph with fewer chains, the number of nodes within each chain is large, which facilitates the learning of connectivity via learning an embedding of the nodes within the chain. As we increase the number of chains while keeping the number of nodes fixed, the length of each chain becomes smaller. Therefore, transformers struggle to learn a vertex-embedding heuristic to learn connectivity for disconnected chain graphs. In Figure 10, when we increase the model size to $d_{emb} = 1024$,

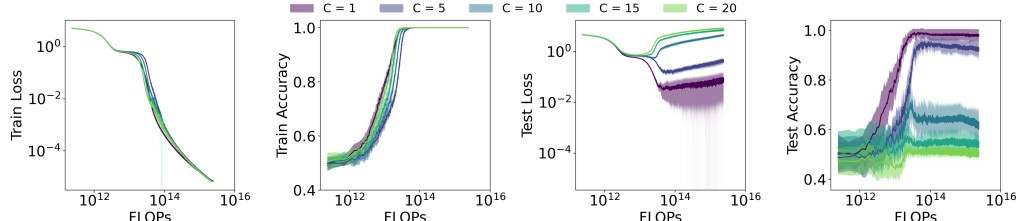

Figure 9: Accuracy and loss vs. training compute on the connectivity task for a disconnected chain graph with various chain sizes with number of chains $C = 1, 5, 10, 15, 20$ and number of nodes $n = 100$ with $d_{emb} = 256$.

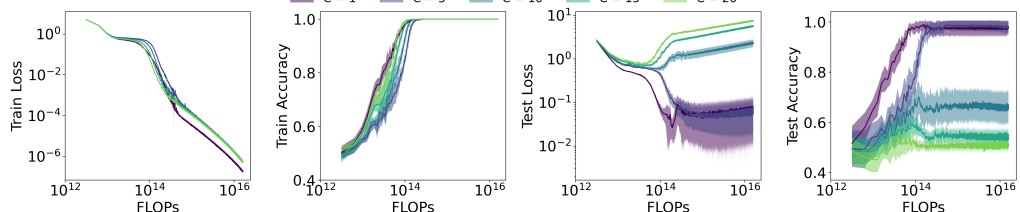

Figure 10: Accuracy and loss vs. training compute on the connectivity task for a disconnected chain graph with various chain sizes with number of chains $C = 1, 5, 10, 15, 20$ and number of nodes $n = 100$ with $d_{emb} = 1024$.

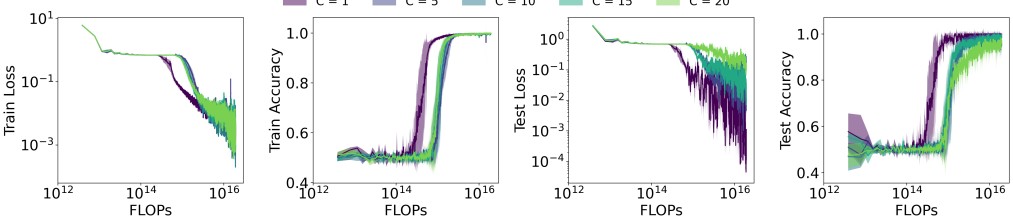

Figure 11: Accuracy and loss vs. training compute on the connectivity task for a disconnected chain graph with various chain sizes with number of chains $C = 1, 5, 10, 15, 20$ and number of nodes $n = 400$ with $d_{emb} = 256$.

while keeping the graph size the same as in Figure 9, we observe that model scaling does not assist transformers to learn the graph connectivity task for the number of chains $C = 10, 15$ and $20$. However, when we increase the graph size to $n = 400$ in Figure 11, we can observe a sharp increase in test accuracy for the number of chains $C = 10, 15, 20$. Increasing the number of nodes in the graph increases the average chain length and provides transformers with enough data to learn the connectivity task even for a larger number of chains, which improves their performance.

## 5 CONCLUSION

In this study, we investigate the reasoning capabilities of transformers by inspecting whether they can reason about transitive relations, and equivalently, whether they can learn the graph connectivity task from the connectivity information between node pairs which are provided as individual training examples. We scale the model dimension and the graph size to observe how it impacts transformers' performance in learning graph connectivity. We observe that model scaling improves the performance of transformers for grid graphs more than the disconnected chain graphs. Moreover, scaling the graph size, transformers' performance remains consistent for grid graphs of different graph sizes and improves their performance for disconnected chain graphs as the graph size increases. In addition, experimental analysis suggests that data scaling benefits transformer performance more than model scaling in learning graph connectivity for higher-dimensional grid graphs or disconnected chain graphs with a larger number of chains.

As our investigation focused on transformers trained on synthetic graphs, one promising direction for future work is to assess the capabilities of transformers to learn connectivity of graphs that arise in real-world data, e.g., knowledge graphs, causal graphs, etc. Further research is needed to determine whether pretrained LLMs have acquired a generalizable node embedding of such real-world graphs.

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

# 6 APPENDIX

## 6.1 GRID GRAPH GENERATION

In this subsection, we present the pseudocode to generate a grid graph from the number of nodes $n$ and grid dimension $k$ in Algorithm 1.

Initially, we compute base $b$ as the celling of the $k$-th root of $n$. We represent each node as a $k$-dimensional vector, which is a number of base $b$ with $k$ digits. First, we add $n$ nodes with index $0$ to $n-1$ to $grid\_graph$. We convert each node index $node\_id$ in range $[0, n-1]$ into a number of base $b$ with $k$ digits. We maintain two different dictionaries $node2vector$ and $vector2node$, for each node and their corresponding vector representation.

Next, for each node $u$ in $grid\_graph$ we obtain the vector representation from the $node2vector$ dictionary. Next, we obtain a new vector representation $new\_vector$ by incrementing each of the $k$ different indexes of node $u$'s vector representation. Next, we check the dictionary $vector2node$ whether there exists a node $v$ with the new vector representation, $new\_vector$. If such a node $v$ exists in the dictionary $vector2node$, then we add an edge $(u, v)$ to the $grid\_graph$.

We return $grid\_graph$ at the end of Algorithm 1.

---

**Algorithm 1:** Procedure to generate a $k$-dimensional grid graph with $n$ nodes.

```
1  function generate_grid_graph(n, k)
2      b ← ⌈n^(1/k)⌉
3      grid_graph ← ∅
       /* Add nodes to the graph */
4      for node_id ← 0 to n − 1 do
5          grid_graph ← grid_graph ∪ {node_id}

       /* Map nodes to k-dimensional vectors */
6      node2vector ← {}
7      vector2node ← {}
8      for node_id ← 0 to n − 1 do
9          vector ← [ ]
10         base ← b
11         num ← node_id
12         for j ← 1 to k do
13             vector.append(num mod b)              /* modulo by base */
14             num ← ⌊num/b⌋
15         node2vector[node_id] ← vector
16         vector2node[vector] ← node_id

       /* Add edges between nodes in grid graph */
17     for u ∈ grid_graph do
18         vector ← node2vector[u]
19         for j ← 0 to k − 1 do
20             new_vector ← vector
21             new_vector[j] ← new_vector[j] + 1
22             if new_vector ∈ vector2node.keys()
23                 v ← vector2node[new_vector]
24                 add_edge(grid_graph, u, v)

25     return grid_graph
```

---

## 6.2 CHAIN GRAPH GENERATION

Next, we provide the pseudocode to generate a chain graph with parameters $total\_nodes$ and $number\_of\_chains$ Algorithm 2. For a given number of $total\_nodes$ and $number\_of\_chains$, we first compute the average chain length, $L$ for each chain. Next, we add nodes with $node\_id$ in range $[1, total\_nodes]$ in the $chain\_graph$.

We add all nodes to a list $available\_nodes$. Next, we sample $L$ nodes from the list $available\_nodes$ without replacement and form a chain by connecting the nodes in the chain nodes. We repeat this process as many times as the parameter $number\_of\_chains$.

After creating chains, if more than one node is remaining in the $available\_nodes$, we connect them to produce another chain.

At the end of the Algorithm 2, we return the $chain\_graph$.

---

**Algorithm 2:** Pseudocode for chain graph generation with parameters $total\_nodes$ and $number\_of\_chains$.

```
1  function generate_chain_graph(total_nodes, number_of_chains)
```
$$2 \quad L \leftarrow \left\lfloor \frac{total\_nodes}{number\_of\_chains} \right\rfloor \qquad \text{/* average chain length per chain */}$$
```
       /* Initialize node set and empty graph */
3      chain_graph ← ∅
4      for node_id ← 1 to total_nodes do
5      ⌊ chain_graph ← chain_graph ∪ {node_id}

6      available_nodes ← [1, 2, . . . , total_nodes]
       /* Create number_of_chains disjoint chains of length L */
7      for c ← 1 to number_of_chains do
8          sampled_nodes ← sample_without_replacement(available_nodes, L)
9          remove chain_nodes from available_nodes
10         for i ← 1 to L − 1 do
11             u ← sampled_nodes[i];   v ← sampled_nodes[i + 1]
12         ⌊ add_edge(chain_graph, u, v)

       /* Wire any leftover nodes into one additional chain */
13     if |available_nodes| > 1 then
14         for i ← 1 to |available_nodes| − 1 do
15             u ← available_nodes[i];   v ← available_nodes[i + 1]
16         ⌊ add_edge(chain_graph, u, v)

17     return chain_graph
```

## 6.3 TRAIN AND TEST SET GENERATION

In algorithm 3, we present the pseudocode to generate a train and test set from a given directed graph. First, based on reachability, we add all positive and negative pairs to $train\_positive\_pairs$ and $train\_negative\_pairs$, and we exclude node pairs with identical node index e.g., $(u, u)$. Next, we sample $M_{test} = 40$ pairs $(u, v)$ such that $dis_{graph}[u][v] > 1$ and add $(u, v)$ to $test\_positive\_pairs$ and the reverse negative pair $(v, u)$ to $test\_negative\_pairs$. Also, we remove $(u, v)$ from $train\_positive\_pairs$ and remove $(v, u)$ to $train\_negative\_pairs$. After that, we compute all possible disconnected negative pairs $(u, v)$ such that $u$ is not reachable from $v$ and $v$ is not reachable from $u$. Lastly, we sample $M_{test}$ pairs $(u, v)$ from disconnected negative pairs and add $\{(u, v), (v, u)\}$ to $test\_negative\_pairs$ removing from $train\_negative\_pairs$.

---

**Algorithm 3:** Generate training and test set from directed graph.

```
1  function generate_train_test_pairs(graph, M_test)
2     dis_graph ← distances from each node using edges in graph
3     train_positive_pairs ← [ ];   train_negative_pairs ← [ ]
4     test_positive_pairs ← [ ];   test_negative_pairs ← [ ]
      /* Populate train sets by reachability */
5     for u ∈ graph do
6        for v ∈ graph do
7           if u = v
8              continue
9           if v ∈ dis_graph[u]
10             train_positive_pairs.append((u, v))
11          else
12             train_negative_pairs.append((u, v))

      /* Move M_test multi-hop positive and reverse negative pairs to test set */
13    for i ← 1 to M_test/2 do
14       (u, v) ← Randomly sample from train_positive_pairs s.t. dis_graph[u][v] > 1
15       test_positive_pairs ← test_positive_pairs ∪ (u, v)
16       train_positive_pairs ← train_positive_pairs \ (u, v)
17       test_negative_pairs ← test_negative_pairs ∪ (v, u)
18       train_negative_pairs ← train_negative_pairs \ (v, u)
      /* Build disconnected negative candidates (no path either direction) */
19    non_reverse_pairs ← []
20    node_list ← []
21    for node ∈ graph do
22       node_list.append(node)
23    l ← |node_list|
24    for i ← 1 to l do
25       for j ← i + 1 to l do
26          u ← node_list[i];   v ← node_list[j]
27          if (v ∉ dis_graph[u]) ∧ (u ∉ dis_graph[v])
28             non_reverse_pairs ← non_reverse_pairs ∪ (u, v)

      /* Sample disconnected negatives into test and remove from train negatives */
29    sampled_non_reverse_pairs ← sample_without_replacement(non_reverse_pairs, M_test)
30    for edge ∈ sampled_non_reverse_pairs do
31       u ← edge[0];   v ← edge[1]
32       test_negative_pairs ← test_negative_pairs ∪ {(u, v), (v, u)}
33       train_negative_pairs ← train_negative_pairs \ {(u, v), (v, u)}
34    return train_positive_pairs, train_negative_pairs, test_positive_pairs, test_negative_pairs
```

## 6.4 TRANSITIVITY AND GRID GRAPH CO-ORDINATES RELATIONSHIP

We provide an example of a 2-dimensional grid graph with 4 nodes and present the relationship between the transitivity function between a start and end node with the difference of their vector representation in Figure 12 and Table 13.

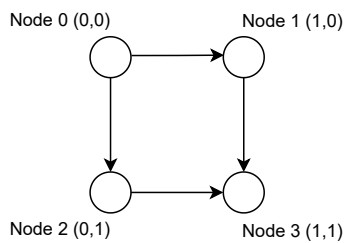

| $start, end$ | $\Psi(end) - \Psi(start)$ | $\mathcal{T}(start, end)$ |
|---|---|---|
| 0, 1 | (1, 0) | 1 |
| 0, 2 | (0, 1) | 1 |
| 0, 3 | (1, 1) | 1 |
| 1, 0 | (-1, 0) | 0 |
| 1, 2 | (-1, 1) | 0 |
| 1, 3 | (0, 1) | 1 |
| 2, 0 | (0, -1) | 0 |
| 2, 1 | (1, -1) | 0 |
| 2, 3 | (1, 0) | 1 |
| 3, 0 | (-1, -1) | 0 |
| 3, 1 | (0, -1) | 0 |
| 3, 2 | (-1, 0) | 0 |

Figure 12: A two-dimensional grid graph with four nodes.

Figure 13: Relation between difference of vectors, $\Psi(end) - \Psi(start)$ and connectivity function $\mathcal{T}(start, end)$ for a two-dimensional grid graph with four nodes.

## 6.5 HARDWARE:

All the experiments are performed on a Linux server with a 2GHz AMD EPYC 7662 64-Core Processor and 1 NVIDIA A100-PCIe GPU with 40GB memory.

