# OpenReview forum: "Transformers Can Learn Connectivity in Some Graphs but Not Others"
_ICLR.cc/2026/Conference — ICLR 2026 Conference Withdrawn Submission_

### Official Review · Reviewer_LQYL · 2025-10-15

**Soundness:** 3
**Presentation:** 3
**Contribution:** 1
**Rating:** 2
**Confidence:** 5

**Summary:**

This paper aims to study the ability of Transformer models to infer transitive relationships between entities from training data. Unlike existing methods that provide necessary details in the prompt, this paper explores the ability of Transformers to infer transitive causal relationships that exist in the training data. This task has many real-world applications. Experiments are performed on different settings to check the ability of transformer models on both grid based and disconnected graphs.

**Strengths:**

1. The paper is written well and easy to understand.
2. The underlying problem being studied i.e., the ability of transformers in identifying the underlying transitive relationships in the training data, is timely and important.

**Weaknesses:**

1. The major weakness of this paper is that, it simply presents a bunch of results with no analysis (both theoretically and empirically). It is not analyzed why the transformer models perform better on grid like structures but not on disconnected graphs.
2. Experiments are performed on purely synthetic datasets and hence it is not clear how the claims transfer to real world scenarios.
3. To claim anything about the ability of LLMs, it is crucial to study state-of-the-art LLMs. However, this paper trains its own transformer models on synthetic data, which is limiting. Further it generalizes its claims to all transformers without rigorous experimentation and analysis (L84-L87).
4. Experiments are conducted on only two graph types (grid and disconnected) and such graphs are not a good proxies for real-world causal graphs.

Minor:
1. While I am not an expert in atmospheric physics / meteorology, it appears that the example of "thunder-lighting" seems incorrect in Line 77. Lightning and thunder can indeed be causally related to each other.

**Questions:**

Please address the points raised in the Weaknesses section.

---

### Official Review · Reviewer_AbFk · 2025-10-24

**Soundness:** 2
**Presentation:** 3
**Contribution:** 2
**Rating:** 2
**Confidence:** 4

**Summary:**

The paper studies whether Transformers can learn transitivity as graph connectivity. Models are trained on synthetic grid and disconnected chain graphs to test how model size, graph size, and graph structure affect learning. Results show Transformers perform well on low-dimensional grid graphs but fail on disconnected ones. Scaling the graph size helps more than scaling the model.

**Strengths:**

Clear motivation connecting transitivity reasoning with graph connectivity.

Systematic experiments across multiple variables (model size, graph size, dimension, components).

Results are consistent and well-visualized.

**Weaknesses:**

The paper lacks novelty. The authors revisit well-known questions without providing new insights or understanding.

The main contributions are as follows:

1. Reformulating reasoning tasks as graph tasks and observing that Transformers struggle with certain graph reasoning patterns—an issue already discussed in [1, 2].

2. Studying the effect of increasing graph dimensions in grid graphs, which parallels prior analyses of node-degree increases in [2].

3. Noting that disconnected nodes are harder to learn, a phenomenon previously explored in [3].

4. Confirming that model scaling still faces inherent limitations, as verified in [2].

In conclusion, the paper largely reiterates established findings on Transformer behaviors without introducing novel ideas or explanations for the underlying mechanisms.


[1] Saparov A, Pawar S A, Pimpalgaonkar S, et al. Transformers Struggle to Learn to Search[C]//The Thirteenth International Conference on Learning Representations.
[2] Bachmann G, Nagarajan V. The Pitfalls of Next-Token Prediction[C]//International Conference on Machine Learning. PMLR, 2024: 2296-2318.
[3] Wang S, Shen Y, Feng S, et al. Alpine: Unveiling the planning capability of autoregressive learning in language models[J]. Advances in neural information processing systems, 2024, 37: 119662-119688.

**Questions:**

See weaknesses. If the authors can clearly state what is novel in their study or how it significantly differs from prior work, I would be willing to raise my score.

---

### Official Review · Reviewer_Ag7e · 2025-10-28

**Soundness:** 2
**Presentation:** 3
**Contribution:** 2
**Rating:** 2
**Confidence:** 3

**Summary:**

The paper investigates whether transformers can learn transitivity from training data and how this ability scales with model size. Unlike prior work that relies on in-context demonstrations, this study examines whether models can internalize the concept of transitivity purely through training. The authors find that the dimensionality of the underlying graph strongly influences model performance, and that transformers struggle particularly with disconnected graphs. While the setup offers a clear and controlled environment to study this reasoning capability, the evaluation remains overly simplistic, lacks a link to real-world relevance, and does not explore architectural modifications that could help models learn such logical rules more effectively. Overall, the paper would benefit from a broader exploration to strengthen its overall impact.

**Strengths:**

1. The paper presents a well-designed experimental setup to examine how transformers generalize transitivity when learned directly from training data. The structured synthetic environment allows for a clear analysis of the models’ failure modes.

2. The scaling analysis is particularly valuable, as it sheds light on how model size, dataset scale, and graph connectivity influence the learning and application of transitivity.

3. The inclusion of grid-graph-like structures extends the scope of prior work and provides a richer framework for assessing transformers’ ability to apply transitivity across more complex relational patterns.

**Weaknesses:**

1. The claim that prior works don’t focus on learning via training is incorrect. Vashishtha et al. 2025 (Teaching Transformers Causal Reasoning through Axiomatic Training), specifically train transformer models from scratch on applying transitivity and evaluate the ability to evaluate it on OOD and complex graphs. As far as I can understand, learning from in-context examples is only used for baselines in that work, whereas models are trained from scratch and fine-tuned on the task of applying transitivity. Can you please confirm this?

2. I appreciate the study’s focus on transformers’ ability to learn transitive relations and the observed difficulty with disconnected graphs. However, before generalizing that transformers inherently exhibit this limitation, the architectural design should be examined more closely. In particular, the role of positional encodings deserves detailed analysis, as prior work such as The Impact of Positional Encoding on Length Generalization in Transformers (Kazemnejad et al., 2023), Positional Description Matters for Transformers Arithmetic (Shen et al., 2023), and Teaching Transformers Causal Reasoning through Axiomatic Training (Vashishtha et al., 2024) has shown that absolute positional encoding often limits generalization. The authors should therefore consider evaluating with Rotary Positional Encodings, which have become the standard in modern architectures and consistently demonstrate stronger generalization across related reasoning tasks.

3. While I appreciate the clarity of the analysis in highlighting specific limitations of transformer models, the work would be significantly stronger if it explored why these failures occur. In particular, the paper could investigate the underlying factors that prevent transformers from learning transitivity in disconnected graphs and propose potential mechanisms or architectural adjustments to address this issue. It would also be valuable to analyze why models perform better on grid graphs compared to disconnected graphs, as understanding this contrast could provide deeper insights into the inductive biases that govern transformer reasoning.

**Questions:**

1. Why did the authors not incorporate an analysis of performance across all positional encodings, given the limitations of absolute PE. I am happy to increase scores if authors can incorporate that comparison.

2. How can current work's finding improve or relate to general reasoning capabilities of language models? How does the current paper's setup connect to what current models lack, and how it could potentially be improved? I would really appreciate this aspect to be explored more heavily since right now the work only focuses on a toy setup with no strong connection to why these findings are important.

3. Why do models perform better on grid graphs but not on disconnect graphs? What potential measures can improve performance on disconnected graphs (this can also help target more real world impact).

---

### Official Review · Reviewer_UmfW · 2025-11-06

**Soundness:** 3
**Presentation:** 3
**Contribution:** 2
**Rating:** 4
**Confidence:** 4

**Summary:**

The paper aims to examine the ability of transformers to comprehend graph connectivity information stored in training data. The paper examines transformers mainly on two graph primitives: (A) Grid Graphs and (B) Disconnected Chain Graphs. The main experimental setup involves in providing a transformer with a pair of node embeddings as input and then predicting if there is a directed path that exists from the first node in the pair to the second. The behaviour of the transformer(s) is studied in different settings: (A) testing performance on grid graph and disconnected chain graphs (B) increasing model size (C) increasing graph size (number of nodes) (D) varying grid graph dimensionality (E) varying number of disconnected components.

**Strengths:**

* Studying graph comprehension in a training setting is novel and useful to the field.
* Experiments are exhaustive and thorough.
* Paper is well-written and all relevant details are sufficiently described.

**Weaknesses:**

* The greatest weakness of this work is its scope, which is extremely limited. The paper motivates itself by stating the importance of transitivity in various reasoning applications but does not perform any experiments that closely match what a reasoning setup might actually look like. This makes this work more of a preliminary investigation into the matter graph comprehension (inferring connectivity and transitive connection in particular) than a more comprehensive look into the area with insights that can directly be used in reasoning domains.
* More concretely, the above concern is reflected in two design choices in the paper: (A) Spatial Node Embeddings: Graphs, in their most general form, do not exist in space. By associating each node with what seems to be an embedding in a k-dimensional space, the paper significantly deviates from what might be expected in a setting closer to reasoning. In a more general setting, there is no analogue for dimensionality, which is used as an important parameter that is varied in the experiments of the paper. (B) The Grid Graph: the grid graph is too restrictive a class of graphs to be of any use in a more general reasoning setting. The connectivity function can be entirely constructed as a program involving subtraction and comparison operations with no need for recall at all. This will not extend well to more general graphs that are more irregular in structure and that do not have embeddings that neatly follow some well-defined principle that governs their connectivity, i.e connectivity cannot be solely inferred from node embeddings.

**Questions:**

* The setup chosen by the authors is extremely restrictive,  is there any motivation as to why this was designed this way to answer the broader question of transitive reasoning in training examples?
* Do the authors think adjusting the number of nodes  and the number of chains (in the case of disconnected chain graphs) while maintaining a constant number of nodes per chain, i.e node density, will result in situations where the performance of the transformer remains constant?
* Can the authors explain how the findings of their paper may relate back to transitive reasoning in a more general setting? That will tie the conclusions back to the motivation of the paper.

---

### Note · Authors · 2025-11-20

**Comment:**

Dear Reviewers,

We decided to withdraw the paper and are thankful for providing the comments. We will try to improve the paper and submit it to a future conference.

Thanks
Authors,
Submission No. 1019,
ICLR 2026

**Withdrawal Confirmation:**

I have read and agree with the venue's withdrawal policy on behalf of myself and my co-authors.